# Paintable Silicone-Based Corrugated Soft Elastomeric Capacitor for Area Strain Sensing

**DOI:** 10.3390/s23136146

**Published:** 2023-07-04

**Authors:** Han Liu, Simon Laflamme, Matthias Kollosche

**Affiliations:** 1Department of Civil, Construction, and Environmental Engineering, Iowa State University, Ames, IA 50011, USA; laflamme@iastate.edu (S.L.); matthias.kollosche@gmail.com (M.K.); 2Department of Electrical and Computer Engineering, Iowa State University, Ames, IA 50011, USA; 3Harvard John A. Paulson School of Engineering and Applied Sciences, Harvard University, Cambridge, MA 02138, USA

**Keywords:** structural health monitoring, large area electronics, soft elastomeric capacitor, stretchable sensor, composite, flexible sensor, silicone, polymer, strain

## Abstract

Recent advances in soft polymer materials have enabled the design of soft machines and devices at multiple scales. Their intrinsic compliance and robust mechanical properties and the potential for a rapid scaling of the production process make them ideal candidates for flexible and stretchable electronics and sensors. Large-area electronics (LAE) made from soft polymer materials that are capable of sustaining large deformations and covering large surfaces and are applicable to complex and irregular surfaces and transducing deformations into readable signals have been explored for structural health monitoring (SHM) applications. The authors have previously proposed and developed an LAE consisting of a corrugated soft elastomeric capacitor (cSEC). The corrugation is used to engineer the directional strain sensitivity by using a thermoplastic styrene-ethylene-butadiene-styrene (SEBS). A key limitation of the SEBS-cSEC technology is the need of an epoxy for reliable bonding of the sensor onto the monitored surface, mainly attributable to the sensor’s fabrication process that comprises a solvent that limits its direct deployment through a painting process. Here, with the objective to produce a paintable cSEC, we study an improved solvent-free fabrication method by using a commercial room-temperature-vulcanizing silicone as the host matrix. The matrix is filled with titania particles to form the dielectric layer, yielding a permittivity of 4.05. Carbon black powder is brushed onto the dielectric and encapsulated with the same silicone to form the conductive stretchable electrodes. The sensor is deployed by directly painting a layer of the silicone onto the monitored surface and then depositing the parallel plate capacitor. The electromechanical behavior of the painted silicone-cSEC was characterized and exhibited good linearity, with an R${}^{2}$ value of 0.9901, a gauge factor of 1.58, and a resolution of 70 με. This resolution compared well with that of the epoxied SEBS-cSEC reported in previous work (25 με). Its performance was compared against that of its more mature version, the SEBS-cSEC, in a network configuration on a cantilever plate subjected to a step-deformation and to free vibrations. Results showed that the performance of the painted silicone-sCEC compared well with that of the SEBS-cSEC, but that the use of a silicone paint instead of an epoxy could be responsible for larger noise and the under-estimation of the dominating frequency by 6.7%, likely attributable to slippage.

## 1. Introduction

Recent advances in soft polymer materials have enabled the design of soft machines and devices at multiple scales. Stretchable and/or flexible sensors that mimic the sensory capabilities of human skin have gained significant attention in the structural health monitoring (SHM) research community [1] because of their significantly improved surface compliance compared to conventional sensors [2,3,4]. These large area electronics (LAE) [5,6] are particularly promising for covering large surfaces and complying with complex and irregular surfaces when deployed as dense sensor networks, thus enabling area sensing. Examples of applications to SHM include conductive polymers [7,8], flexible circuit boards and sheets [9,10,11], and flexible piezoelectric sensor networks [12,13].

The authors have recently proposed a large-area stretchable strain gauge based on a surface-corrugated soft elastomeric capacitor (cSEC) that transduces strain into a measurable change in capacitance, with a reported minimum resolution of 25 με [14]. The cSEC technology has demonstrated success in many applications, including fatigue crack monitoring [15], concrete crack detection [16], angular motion sensing [17], and biomechanical sensing [18]. The cSEC is fabricated from a styrene-ethylene-butylene-styrene (SEBS) block copolymer matrix doped with 15% vol titania (TiO${}_{2}$) to form the dielectric layer with a permittivity of 5.56 [14], thus improving its strain sensitivity, and doped with 15% vol conductive carbon black particles to form the electrodes. Details on the fabrication process are reported in the literature [19]. Importantly, the sensor fabrication requires toluene to dissolve the SEBS particles prior to mixing the titania particles. The presence of toluene in the solution has known negative environmental effects [20], and hinders direct deposition of the solution onto a surface of interest because of possible swelling and dissolution of the SEBS-based electrodes. Instead, the sensors are fabricated in a laboratory environment and adhered onto surfaces using an epoxy resin, which is commonly associated with sensors’ deployment, where cyanoacrylate [21], acrylic adhesive [22], hot melt adhesive [23], and fibroin adhesive [24] have been applied as the bonding agent.

The objective of this paper is to study a paintable version of the cSEC. An improved solvent-free fabrication process is attained using a commercial room-temperature-vulcanizing silicone. In comparison to the thermoplastic elastomers (e.g., SEBS), silicone is chemical cross-linked and is constituted from siloxane that confers the typical rubbery property with excellent stretchability, bio-compatibility, heat and chemical resistance, physiological inertia, and hydrophobicity [25,26]. The degradation of silicone materials under different environmental and storage conditions was studied in numerous works that validated its durability [27,28,29]. Therefore, silicone is frequently employed as a host matrix to integrate micro- and nano- particles to produce coating materials and stretchable substrates that are used to fabricate sensors. Specific examples of silicone-based sensors include a 3D-printed surface-doped porous wearable sensor fabricated by embedding graphene onto the silicone surface [30], a soft force sensor fabricated by blending magnetic powder with silicone [31], a highly sensitive piezoresistive sensor produced by mixing chopped carbon fibers and silicone elastomers into an auxetic structure shape [32], an ultra-robust wide-range pressure sensor constructed by coating polyurethane foam with a silicone sheet and carbon nanotube-dispersed thermoplastic polyurethane ink [33], and an implantable strain sensor fabricated by intertwining organogel fiber and a silicone fiber in a double helix structure [34].

The sensing performance of a silicone-based sensor is strongly dependent upon its stiffness and consequently upon the cross-link density of the silicone and its permittivity [35]. Thus, the elastomers forming the dielectric are preferably doped with high permittivity nanoparticles. Particularly, fillers such as titania [36,37], lead magnesium niobate [38], montmorillonite [39], ZnO nanoparticles [40], and BaTiO${}_{3}$ particles [41] have been used to enhance the relative permittivity and thermal stability of silicone. Here, the silicone matrix is filled with 3 wt% PDMS-coated titania particles to form the dielectric, yielding a permittivity of 4.05.

The electrodes of the silicone-based SEC, here termed silicone-cSEC to distinguish from the SEBS-cSEC, is fabricated by brushing carbon black powder onto both sides of the dielectric to form electrodes, and the carbon black powder is encapsulated. A thin layer of silicone is directly painted onto the monitored surface to bond the pre-fabricated silicone-cSEC. This configuration enables a polymer-on-polymer contact that can use interfacial chain entanglements to provide sufficient bonding strength for direct deployment [42]. The sensing properties of the silicone-cSEC are characterized through a compression–tension bending test conducted on a fiberglass cantilever beam. The performance of the bonding effects is compared against that of the SEBS-cSEC using a traditional epoxy bonding method through both a quasi-static and a dynamic test.

The rest of the paper is organized as follows. Section 2 provides the background on silicone-cSEC technology, including its fabrication process and the derivation of the electromechanical model. Section 3 presents and describes experimental configurations and procedures. Section 4 presents and discusses results, starting with the study of its electromechanical behavior, followed by an analysis of its performance in a network configuration. Section 5 concludes the paper.

## 2. Background

This section provides the necessary background on the silicone-cSEC technology, which includes its fabrication process and the derivation of the electromechanical model for in-plane strain sensing.

### 2.1. Fabrication Process

The silicone-cSEC is a highly flexible, stretchable, and scalable thin-film strain sensor composed of a dielectric layer sandwiched between two conductive electrodes. The fabrication process is illustrated in Figure 1.

The commercially available liquid silicone Wacker 7670 A and B components are used as the host matrix. Both the A and B components contain siloxane and silica particles in the range of 30 to 40 wt% [43]. The silicone has a relative permittivity of 2.9 [44], is elastically stretchable beyond 200% [45], and has a stiffness of approximately 220 kPa [45] in its pure form. The relative permittivity is boosted using high permittivity titania particles dispersed in the pre-polymer before the A and B components are mixed and cured. Detail of the fabrication process (Figure 1) is as follows.

PDMS-coated titania ${\mathrm{TiO}}_{2}$(−OSI(CH${}_{3}$)${}_{2}-$) (TPL, Inc., Albuquerque, NM) particles with an average diameter of 100 nm are added in 3 wt% to 5 mL of the liquid silicone WACKER Elastosil P 7670 A (Polydimethyl siloxane (63148-62-9), Polydimethylsiloxane vinyl terminated (68083-19-2), (TSRN 38673700-5112 P)) for a concentration of 30 g/L.Rutile titania particles are uniformly dispersed in the silicone matrix using a low-speed homogenizer for 600 s at 650 RPM while the solution is cooled in an iced water bath.A volume of 5 mL of the liquid silicone WACKER Elastosil P 7670 B (Polydimethyl siloxane (63148-62-9), (TSRN 38673700-5101P), Polydimethylsiloxane vinyl terminated (68083-19-2), Silazanetreated Silica (68909-20-6), Polydimethyl hydrogenmethyl siloxane (69013-23-6)) is added into the stock solution and mixed using a shear mixer for 180 s at 2000 RPM, yielding a dynamic viscosity of approximately 2000 cP.The resulting silicone–titania solution is drop-cast onto an 80 mm × 80 mm non-stick square steel mold. The steel mold contains grooves to create a corrugated pattern. The use of surface corrugation is known to improve strain sensing performance by adding the in-plane stiffness and decreasing the transverse Poisson’s ratio [19].The drop-casted solution is cured under room temperature over 6 h, and the film is subsequently peeled from the mold. The resulting film has a mean thickness of 0.4 mm over the non-corrugated area and a corrugation height of 0.35 mm. Remark that the thickness of the dielectric layer can be tuned by controlling the volume of the silicone–titania solution drop-casted into the steel mold.Carbon black particles are stored in an oven at 50 °C for 24 h to remove moisture, and an anti-static gun (Milty 5036694022153 Zerostat 3) is used to remove the static charge on the surface of the cured dielectric film before brushing electrodes. A dry stacking process through stamping that has no solvent–elastomer interactions [45] can be employed for future studies.The dry carbon black particles are brushed onto both sides of the dielectric layer to form conductive soft stretchable electrodes. The painting process is stopped once the electrode has reached a sheet resistance of approximately 3.6 kΩ/Sq (Botron digital surface resistivity meter, SKU: B8563). In prior work on the SEBS version of the sensor, we conducted accelerating ageing tests and found that the use of carbon black conferred the polymer with long-term durability both mechanically and electromechanically [46]. While a similar study on silicone-cSECs is left to future work, it is hypothesized that the use of carbon black would yield similar conclusions.Adhesive copper tapes are glued on the brushed conductive electrodes to create electrical connections to the data acquisition system (DAQ). A thin layer of PELCO conductive carbon glue (TED Pella, USA) is added to the exposed parts of copper tapes to enhance mechanical bonding strength and minimize signal noise. The resulting silicone-cSEC has a permittivity of 4.05 at 100 Hz (Equation (Equation 1)) with an effective thickness of 0.52 mm for the electrode section of the sensor, which corresponds to an increase of approximately 40% compared to the pure silicone. The Young’s modulus of the cured silicone composite was found to be 305 kPa using a tensile tester under a strain rate of 2.5%/s.As an optional step for deployment, a small amount of WACKER Elastosil P 7670 A and B components are mixed with a weight ratio of 1:1 and applied as a protecting layer onto the surface of the electrodes to improve resilience with respect to weathering. The resulting silicone-cSEC has an initial capacitance of approximately 170 to 200 pF under 1 kHz measuring frequency.

### 2.2. Electromechanical Model

Figure 2A presents a silicone-cSEC with a reinforced diagrid pattern. The strain sensing principle that relates a change in area (i.e., strain) of the sensor (i.e., provoked by strain in the monitored surface) to a measurable change in capacitance can be derived as follows. Take the initial capacitance ($C_{0}$) of a non-lossy parallel plate capacitor:(1)$$C_{0}=e_{0}e_{r}\frac{A}{h}$$
where $e_{0}=8.854$ pF/m is the vacuum permittivity, $e_{r}$ is the polymer’s relative permittivity, A=l·w is the electrode area of length *l* and width *w*, and *h* is the thickness of the dielectric (as annotated in Figure 2B). Assuming small strains along the *x*-direction, the relative change in capacitance $\Delta C/C_{0}$ can be obtained by differentiating Equation (Equation 1)
(2)$$\frac{\Delta C}{C_{0}}=\left(\frac{\Delta l}{l_{0}}+\frac{\Delta w}{w_{0}}-\frac{\Delta h}{h_{0}}\right)=\varepsilon_{x}+\varepsilon_{y}-\varepsilon_{z}$$

Under plane-stress condition and applying Hooke’s Law, $\varepsilon_{z}=-\nu /\left(1-\nu \right)\cdot \left(\varepsilon_{x}+\varepsilon_{y}\right)$, the change in capacitance as a function of surface strain can be written as:(3)$$\frac{\Delta C}{C_{0}}=\frac{1}{1-\nu_{0}}\left(\varepsilon_{x}+\varepsilon_{y}\right)=\lambda_{0}\left(\varepsilon_{x}+\varepsilon_{y}\right)$$
where $\nu_{0}$ is the Poisson’s ratio of the silicone and $\lambda_{0}$ is the gauge factor. Adding surface corrugation to the dielectric layer alters the in-plane stiffness and produces an orthotropic transverse Poission’s ratio $\nu_{xy}=-\varepsilon_{y}/\varepsilon_{x}$. Thus, Equation (Equation 3) becomes:(4)$$\frac{\Delta C}{C_{0}}=\frac{1-\nu_{xy}}{1-\nu }\varepsilon_{x}=\lambda \varepsilon_{x}$$

Equation (Equation 4) can be specialized for a composite configuration where the transverse Poisson’s ratio is modified due to the composite effect with the monitored materials the sensor is adhered onto:(5)$$\nu_{xy,c}=\frac{a\nu_{xy}+b\nu_{m}}{a+b}$$
where $\nu_{m}$ is the Poisson’s ratio of the monitored material and *a* and *b* are weight coefficients representing the composite effect in which a+b=1, depending on the level of adhesion and material stiffnesses. The resulting gauge factor under composite effect is given by:(6)$$\lambda =\frac{1-\nu_{xy,c}}{1-\nu }$$

## 3. Experiments

This section describes the experimental procedures applied in this research. First, the sensing properties of the silicone-cSEC bonded with a silicone layer are characterized. Second, the sensing performance of the silicone-cSEC is compared with that of the SEBS-cSEC in a network configuration.

### 3.1. Cantilever Plate

The sensing properties of silicone-cSEC in terms of the strain sensitivity, signal linearity, and resolution were characterized on a fiberglass cantilever plate subjected to bending. The overall experimental setup is presented in Figure 3A. One end of the fiberglass plate (Garolite G-10/FR4) with geometry of l×w×h=83×142×2.6 mm${}^{3}$ was restrained in the vertical direction by using two clamps. The surface of the fiberglass plate was sanded by subsequently using 400 and 1000 grit sandpaper and cleaned with a fiberglass solvent wash (Interlux solvent 202) to create a bonding area for the sensor. A portion of uncured Wacker 7670 in a 1:1 ratio was painted as a thin layer over the bonding area, and a single silicone-cSEC was deposited over the uncured silicone layer before letting it cure for 6 h, forming the painted silicone-cSEC. As shown in Figure 3B, a resistive strain gauge (RSG) (TML FLA-10-350-11-1LJCT, SGC-28, nominal resistance of 350 ± 1.0 Ω, gauge length of 10 mm) was installed 2 mm away from the right-hand side of the sensor using a non-conductive strain gauge adhesive (CN Cyanoacrylate) to benchmark results.

A quasi-static test was conducted by slowly pushing up and down the free end of the cantilever plate to generate tensile and compressive bending strains. Custom-built DAQ systems (annotated cSEC DAQ in Figure 3A) fabricated with a 24-bit capacitance-to-digital converter multiplexed over 4 channels were used to collect data measured from the cSECs sampled at 40 samples/second (S/s). Actively shielded coaxial cables were used to connect cSECs to the DAQs to remove parasitic capacitance caused by cable connection. Data from the RSG were recorded using a National Instrument 24-bit 350 Ω 3/4 bridge analog input module (NI-9236) sampled at 1000 S/s. Both DAQs were simultaneously operated in LabVIEW. No filtering was applied to the signals.

### 3.2. Sensor Network Configuration

To test the performance, the painted silicone-cSECs were assessed in a network configuration. To evaluate the effect of direct deposition via painting, results were compared against those of an SEBS-cSEC adhered using an epoxy (“epoxied SEBS-cSEC”) and of an silicone-cSEC adhered using the same painted silicone layer (“epoxied silicone-cSEC”). The SEBS-cSECs used in this test were fabricated by following the procedure reported in [19], and their initial capacitance was kept between 220 and 260 pF under 1 kHz measuring frequency. Tests were performed on another cantilever fiberglass plate (*l* × *w*
× h = 813 × 406 × 2.6 mm${}^{3}$) with geometry identical to the one used in the previous test. The sensor network comprised 6 sensors of each type, for a total of 18 cSECs. A total of 14 RSGs were deployed onto the surface of the plate to verify results. The layout of the sensor network showing each sensor type along with their associated bonding method is illustrated in Figure 3D. Each type of cSEC is deployed in-line perpendicular to the fixed support (along the *y*-direction of the fiberglass plate). A bi-component epoxy (JB Weld) is used to adhere the epoxied SEBS-cSEC. Six RSGs were evenly deployed in-line between lines 1 and 2 to measure strain along the *y*-axis, another six RSGs were deployed 5 mm above each cSEC along line 2 to measure strain along the *x*-direction, and two additional RSGs were deployed near the fixed support to map strain deformations around the fixed boundary condition.

Experiments consisted of subjecting the plate to a quasi-static load and free vibrations. As illustrated in Figure 3E, a steel block was place 50.8 mm (2 inches) below the free end of the plate to serve as a fixed stop point and permit repeatability of the tests. A quasi-static test was conducted by manually bending the free end until the deformation reached the steel block, maintaining the deformation for 5 s, and bringing the plate back to its original position. The free vibrations were generated by pushing the plate down to the steel block and releasing it. Both tests were conducted over three times, for a total of six tests, to investigate the repeatability of results. Data were collected using the same setup as for the single silicone-cSEC, but with five custom-built DAQs to record data from the cSECs and two National Instruments modules (NI-9236) to record data from the RSGs (Figure 3F).

## 4. Results and Discussion

This section presents and discusses the experimental results. First, the electro-mechanical behavior of a single painted silicone-cSEC is characterized. Second, the sensing performance of the painted silicone-cSEC in a network configuration is compared against that of a mature epoxied SEBS-SEC, and the effect of direct adhesion is assessed by comparing performance against a silicone-cSEC epoxied onto the surface (“epoxied silicone-cSEC”).

### 4.1. Electro-Mechanical Behavior

Figure 4A presents a time series plot of the raw data measured from a single silicone-cSEC, showing the relative change in capacitance $\Delta C/C_{0}$ and the applied strain measured from the RSG. A close match between the painted silicone-cSEC and RSG signals can be observed. The resulting root mean square error (RMSE) and mean absolute percentage error (MAPE) of the fit are 5.98% and 6.08%, respectively.

Figure 4B plots the relative change in capacitance $\Delta C/C_{0}$ versus the applied strain using data presented in Figure 4A. A linear fit (red solid line) along with the resulting 95% confidence interval bound (dashed green lines) are also shown. The sensor exhibits good linearity within the studied range (−472 με to 580 με), with a goodness-of-fit R${}^{2}$ of 0.9901. The gauge factor under the composite effect is λ=1.58, obtained from the slope of the linear fit. The 95% confidence interval shows an accuracy of 70 με. This accuracy compares well with the reported 25 με resolution for the more mature version of the cSEC (epoxied SEBS-cSEC, [19]). While the environmental effects, such as changes in humidity and temperature, are left to future work, prior work on signal processing algorithms has shown that these effects can be filtered out either through proper algorithmic formulation or through implementation in a Wheatstone bridge configuration [47].

### 4.2. Network Configuration

Figure 5 plots the sensors’ time series signals transformed into strain measurements using the gauge factor λ of 1.58 for both the painted and epoxied silicone-cSECs, and the gauge factor λ=1.46 for the epoxied SEBS-cSEC computed using Equation (Equation 6) with $\nu_{xy,c}=0.27$, in response to a step deformation. There is a general good agreement between all the cSEC’s signals and the RSG’s. The three RSGs transversely deployed in row 1 confirmed that the additive strain among a given row does not vary significantly. The recovery of each signal after the step deformation is difficult to quantify because of the level of the noise, but they generally appear to temporally follow the strain of the RSG. However, the signal of the painted silicone-sSEC is more noisy, with fluctuations remaining approximately within the 70 με bound established above, as is clearly observable under lower strain values (e.g., row 6). Remark that, compared with the individual configuration, additional noise can be attributable to the network configuration that may cause electro-magnetic noise.

Figure 6 is a bar chart of the averaged peak strain amplitudes measured by each sensor deployed from row 1 to row 6 for the step-deformation test. Results compare across sensor types, and error bars indicate the range of the minimum and maximum values measured over the three independent tests. Results show a general agreement between each type of cSEC and the RSG, with the epoxied SEBS-cSEC outperforming other types of cSECs.

Figure 7 shows plots similar to Figure 5, but for the plate subjected to free vibrations. Results are similar to those obtained from the step deformation, whereas there is a general good agreement between signals, but the painted silicone-cSEC exhibits more noise and lower resolution. One can note that the measured strain at rows 5 and 6 are close to the 70 με resolution shown by the orange dashed line. A lag of approximately 0.1 s to 0.2 s was consistently observed for the painted silicone-cSEC (black line), which could be attributed to slippage at the bonding interface caused by the insufficient adhesion of the hydrosilylation silicone to the thermoplastic substrate [48].

Figure 8 compares the peak strain amplitudes along with their exponential fits for rows 1 (Figure 8A) and 4 (Figure 8B) to investigate the measured damping. Rows 5 and 6 were not investigated because of the high level of noise. Once can observe similar trends among all sensors and no decay in performance between rows 1 (higher signal-to-noise ratio) and 4 (lower signal-to-noise ratio).

Figure 9 compares the frequency spectra of the sensors’ signals obtained through a fast Fourier transform (FFT) conducted on the raw time series measurements presented in Figure 7. The dominating frequencies of the painted silicone-cSEC, epoxied SEBS-cSEC, epoxied silicone-cSEC, and RSG are respectively listed in the upper right corner from top to bottom. A dominating frequency of 2.54 Hz corresponding to that of the plate is clearly observed from the RSG. The epoxied versions of the cSEC closely measure the same dominating frequency, with the epoxied SEBS-cSEC being the only cSEC type capable of measuring the dominating frequency in row 6. The signals of the silicone SECs do not raise above noise. Another notable feature is that the painted silicone-cSEC measures a frequency consistently 6.7% lower than that of the RSG. This can be attributed to slippage of the sensor and its hysteresis behavior.

The SNR for both the step-deformation (Figure 10A) and free vibration (Figure 10B) tests are evaluated in Figure 10. The figure presents the SNR values averaged over the three tests, with the error bars showing the range of the minimum and maximum values. It can be observed that the RSG exhibits significantly better SNR values, as expected given the technology’s maturity, followed by the epoxied SEBS-cSEC, the epoxied silicone-cSEC, and the painted silicone-cSEC, except for row 6 under the step-load test. From these results, it appears that the use of an epoxy versus direct deposition results in a higher SNR, attributable to the better sensor-monitored material interface bonding. It can be also observed that the SNR values consistently decrease as the sensors become closer to the plate’s tip and experience smaller strain. This is consistent with the previous observations.

## 5. Conclusions

This paper presented a silicone-based corrugated soft elastomeric capacitor (silicone-cSEC) that can be directly deposited onto a monitored surface. The paintable sensor was achieved by formulating a solvent-free fabrication method using a commercial room-temperature-vulcanizing silicone. The silicone-cSEC was presented as an alternative to a more mature type of cSEC based on a thermoplastic styrene-ethylene-butadiene-styrene (SEBS) block co-polymer that required epoxied SEBS-cSEC on the monitored surface because of the use of solvents in its fabrication process.

The silicone-cSEC was fabricated by dispersing TiO${}_{2}$ in the silicone matrix to increase the permittivity to 4.05 and improve its strain sensitivity. Its sensing properties in terms of the linearity, gauge factor, and resolution were characterized through a quasi-static bending test. The painted silicone-cSEC exhibited good linearity with an R${}^{2}$ value of 0.9901, a gauge factor of 1.58, and a resolution of 70 με. This resolution compared well with that of the epoxied SEBS-cSEC reported in previous work.

A sensor network was constructed by deploying a 3 columns × 6 rows grid array of cSECs onto a cantilever fiberglass plate subjected to a step deformation and to free vibrations. Each column of sensors corresponded to a different type of cSEC. In this experiment, the performance of silicone-cSECs was compared against that of epoxied SEBS-cSECs and painted SEBS-cSEC. The plate was also equipped with off-the-shelf resistive strain gauges (RSGs) to benchmark results.

Results from the experiments showed that the painted silicone-cSECs (1) exhibited good agreement with other sensors in measuring the step-deformation; (2) similar to other cSECs, had a noisy signal under low strain (row 6) that could be attributed to electromagnetic noise caused by the network configuration; (3) were capable of tracking strain under free vibrations, but also exhibited a noisy signal at low strains, (4) had a signal-to-noise ratio lower than the other cSECs, even compared with the epoxied silicone-cSEC; and (5) were the only sensor exhibiting a shift in the measured dominating frequency (6.7%). It can be concluded from these results that the silicone-cSEC technology compares well with the SEBS-cSEC given its relatively early stage of development and that the direct deposition process can be responsible for lower strain sensing performance. These results can be used to develop hybrid flexible electronics with painted silicone serving as the backbone.

Overall, this study demonstrated that an SEC can be successfully fabricated using a solvent-free process and can be used for strain monitoring through direct deposition. Future work is to include the study of the bonding strength between silicone and thermoplastic substrate by using an adhesion promoter or adding a polyurathane-modified layer for the development of test methods to allow the tuning and testing of the adhesion strength.

## Figures and Tables

**Figure 1 sensors-23-06146-f001:**
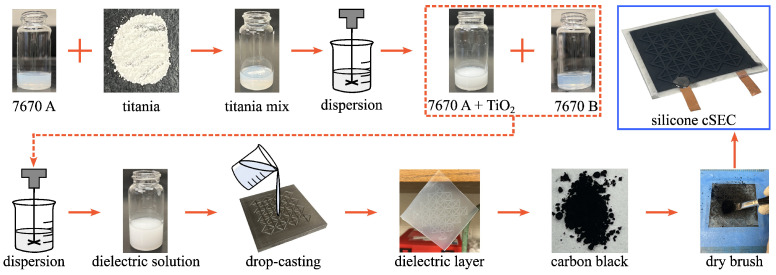
Fabrication process of a silicone-cSEC.

**Figure 2 sensors-23-06146-f002:**
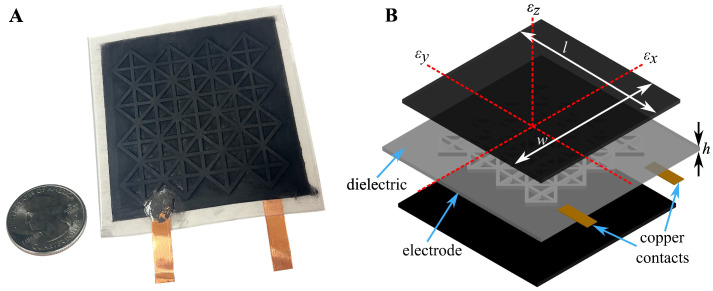
(**A**) Picture of a 76 mm × 76 mm (l×w) silicone-cSEC with a reinforced diagrid pattern; and (**B**) schematic showing the parallel plate capacitor structure with key components and reference axes annotated.

**Figure 3 sensors-23-06146-f003:**
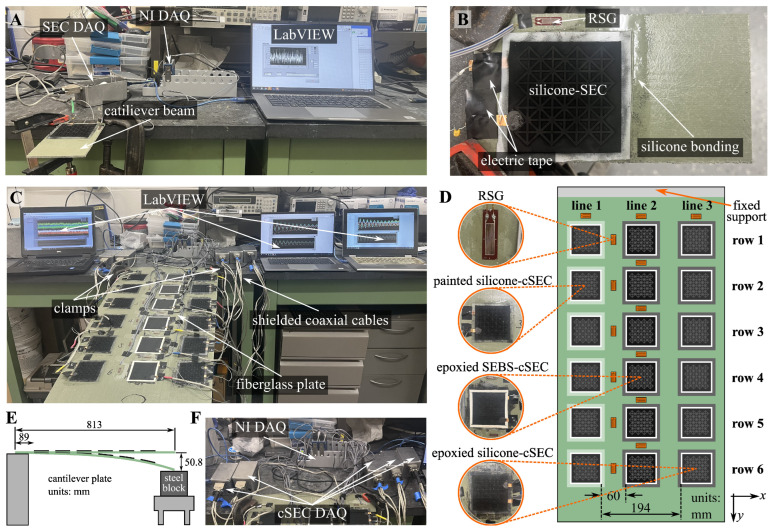
(**A**) Overall experimental setup to characterize a single painted silicone-cSEC; (**B**) closeup view of the painted silicone-cSEC; (**C**) overall experimental setup of the sensor network configuration; (**D**) schematic showing the configuration of the sensor network and the closeup views of the deployed sensors; (**E**) schematic showing the elevation view of the cantilever plate; (**F**) setup of the DAQs used for cSECs and RSGs.

**Figure 4 sensors-23-06146-f004:**
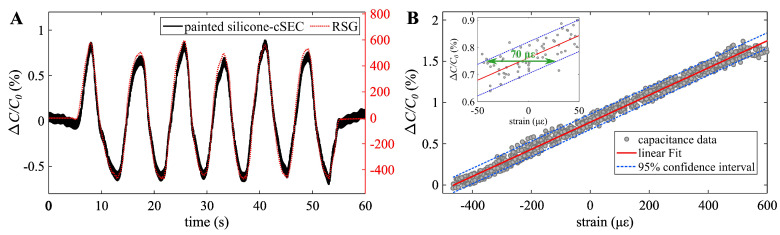
(**A**) Bending test results; and (**B**) relative change in capacitance versus strain along with the 95% confidence interval bound on the fit.

**Figure 5 sensors-23-06146-f005:**
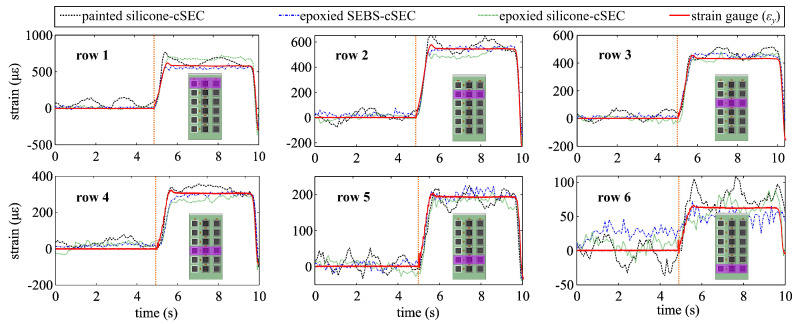
Comparison of the time series measurements under a step deformation.

**Figure 6 sensors-23-06146-f006:**
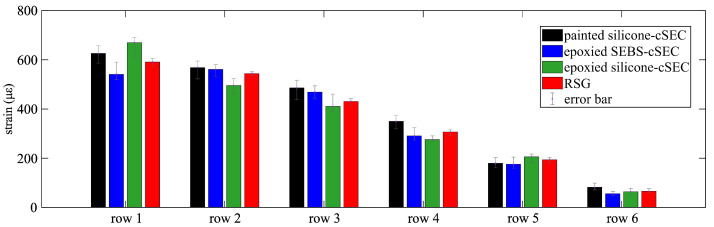
Bar chart comparing the averaged measured peak strain amplitudes for the step-deformation tests.

**Figure 7 sensors-23-06146-f007:**
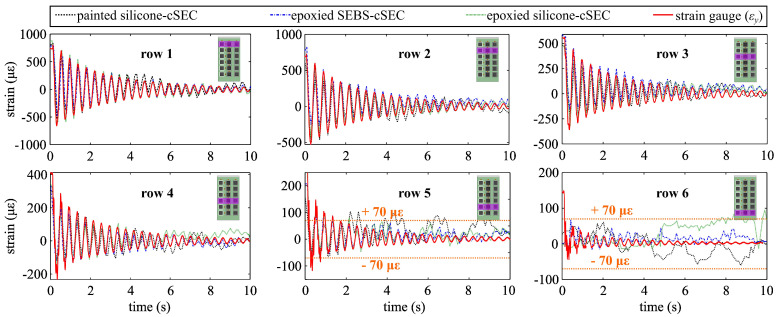
Comparison of the time series measurements under free vibration.

**Figure 8 sensors-23-06146-f008:**
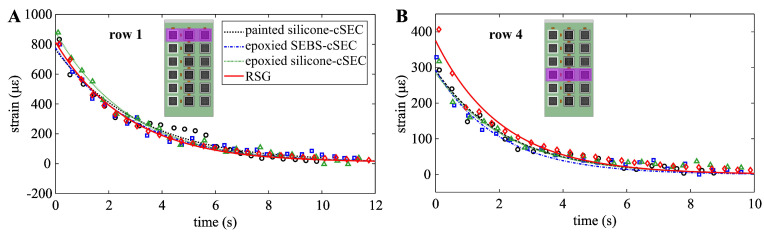
Comparison of measured peak strain amplitudes: (**A**) row 1; and (**B**) row 4.

**Figure 9 sensors-23-06146-f009:**
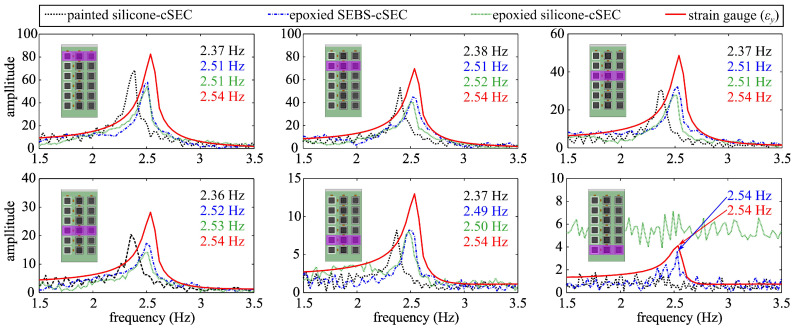
Comparison of the frequency spectra under free vibration.

**Figure 10 sensors-23-06146-f010:**
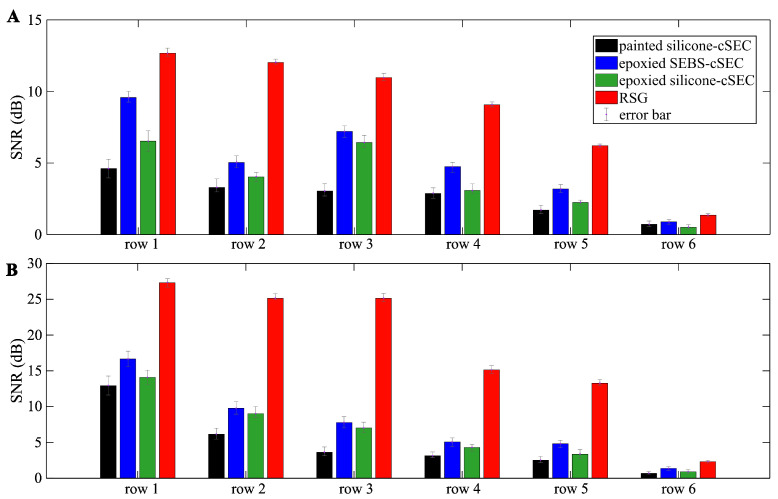
Bar chart comparing the SNR of each sensor across row 1 to row 6 measured under: (**A**) step-deformation test; and (**B**) free vibration test.

## Data Availability

The data presented in this study are available on request from the corresponding author.

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
