# Peer review of "Paintable Silicone-Based Corrugated Soft Elastomeric Capacitor for Area Strain Sensing"

_sensors, 2023, doi:10.3390/s23136146_

Round 1

Reviewer 1 Report

1. How did the authors account for the difference in strain as the sensors are being mounted at different locations of the cantilever plate?

2. In figure 7, the author showed the free vibration data. Is it possible to fit the peak strain of the data to extract a time constant for vibration recovery?

There is a typo in the x-axis label of figure 6. Instead of "raw 6" it should be "row 6".

Reviewer 2 Report

The authors investigated an improved solvent-free fabrication method by using a commercial room-temperature-vulcanizing silicone to produce a paintable cSEC. Generally, the manuscript is well prepared. The following issues should be strengthened to improve the manuscript.

1. For SHM, the durability of sensors is important. How about the durability paintable cSEC?

2. In Fig. 5, great fluctuation of the strain measured by painted silicone-cSEC can be observed, and it does not look like general noise. Please explain.

3. In Fig. 7, the painted silicone-cSEC does not perform well in row 5 and row 6. Please explain.

4. In Fig. 8, the resonance frequency obtained by painted silicone-cSEC is always less than those obtained by other sensors. Please explain.

5. Is there any requirement for the monitored surface?

Reviewer 3 Report

It would be interesting if the authors used advanced techniques such as 3D printing as they could result in a more uniform dielectric layer and a more homogeneous distribution of titania particles, possibly improving sensor performance.

Reviewer 4 Report

This manuscript presents the fabrication of silicone-based corrugated soft elastomeric capacitor and its strain sensing properties. The topic is quite interesting to broad readership. The manuscript is well written. The discussion and conclusion are sound and supported by the results. In my opinion, it can be accepted for publication after revision/clarification as suggested below;

- Based on Figure 5, the response and recovery times should be investigated and discussed. 

- The effect of humidity should be investigated and discussed. 

- Limit of strain detection should be calculated.

- Why a frequency of 2.54 Hz gives the highest amplitude? Please give a reason.

-

Round 2

Reviewer 4 Report

I have no further comments and suggestions for this manuscript.

Minor editing of English language required, i.e., storage conditions "were study" in numerous works....